



# Towards low-cost and high-performance air pollution measurements using machine learning calibration techniques

Peer Nowack[1,2,3,4], Lev Konstantinovskiy[5], Hannah Gardiner[5], and John Cant[5]

[1]Grantham Institute - Climate Change and the Environment, Imperial College London, London SW7 2AZ, UK
[2]Department of Physics, Imperial College London, London SW7 2AZ, UK
[3]Data Science Institute, Imperial College London, London SW7 2AZ, UK
[4]School of Environmental Sciences, University of East Anglia, Norwich NR4 7TJ, UK
[5]Air Public Ltd, London, UK

**Correspondence:** p.nowack@uea.ac.uk

**Abstract.** Air pollution is a key public health issue in urban areas worldwide. The development of low-cost air pollution sensors is consequently a major research priority. However, low-cost sensors often fail to attain sufficient measurement performance compared to state-of-the-art measurement stations, and typically require calibration procedures in expensive laboratory settings. As a result, there has been much debate about calibration techniques that could make their performance more reliable, while

also developing calibration procedures that can be carried out without access to advanced laboratories. One repeatedly proposed strategy is low-cost sensor calibration through co-location with public measurement stations. The idea is that, using a regression function, the low-cost sensor signals can be calibrated against the station reference signal, to be then deployed separately with performances similar to the original stations. Here we test the idea of using machine learning algorithms for such regression tasks using hourly-averaged co-location data for nitrogen dioxide ($NO_2$) and particulate matter of particle sizes smaller than

10 $\mu$m (PM10) at three different locations in the urban area of London, UK. Specifically, we compare the performance of Ridge regression, a linear statistical learning algorithm, to two non-linear algorithms in the form of Random Forest (RF) regression and Gaussian Process regression (GPR). We further benchmark the performance of all three machine learning methods to the more common Multiple Linear Regression (MLR). We obtain very good out-of-sample $R^2$-scores (coefficient of determination) > 0.7, frequently exceeding 0.8, for the machine learning calibrated low-cost sensors. In contrast, the performance of MLR is

more dependent on random variations in the sensor hardware and co-located signals, and is also more sensitive to the length of the co-location period. We find that, subject to certain conditions, GPR is typically the best performing method in our calibration setting, followed by Ridge regression and RF regression. However, we also highlight several key limitations of the machine learning methods, which will be crucial to consider in any co-location calibration. In particular, none of the methods is able to extrapolate to pollution levels well outside those encountered at training stage. Ultimately, this is one of the key

limiting factors when sensors are deployed away from the co-location site itself. Consequently, we find that the linear Ridge method, which best mitigates such extrapolation effects, is typically performing as good as, or even better, than GPR after sensor re-location. Overall, our results highlight the potential of co-location methods paired with machine learning calibration techniques to reduce costs of air pollution measurements, subject to careful consideration of the co-location training conditions, the choice of calibration variables, and the features of the calibration algorithm.





## 1   Introduction

Air pollutants such as nitrogen dioxide ($NO_2$) and particulate matter (PM) have harmful impacts on human health, the ecosystem, and public infrastructure (European Environment Agency, 2019). Moving towards reliable and high-density air pollution measurements is consequently of prime importance. The development of new low-cost sensors, hand in hand with novel sensor calibration methods, has been at the forefront of current research efforts in this discipline (e.g. Mead et al., 2013; Moltchanov

et al., 2015; Lewis et al., 2018; Zimmerman et al., 2018; Sadighi et al., 2018; Tanzer et al., 2019; Eilenberg et al., 2020; Sayahi et al., 2020). Here we present insights from a case study using low-cost air pollution sensors for measurements at three separate locations in the urban area of London, UK. Our focus is on testing the advantages and disadvantages of machine learning calibration techniques for low-cost $NO_2$ and PM10 sensors. The principal idea is to calibrate the sensors through co-location with established high-performance air pollution measurement stations (Fig. 1). Such calibration techniques, if suc-

cessful, could complement more expensive laboratory-based calibration approaches, thereby further reducing the costs of the overall measurement process (e.g. Spinelle et al., 2015; Zimmerman et al., 2018; Munir et al., 2019). For the sensor calibration, we compare three machine learning regression techniques in the form of Ridge regression, Random Forest (RF) regression, and Gaussian Process regression (GPR), and contrast the results to those obtained with standard Multiple Linear Regression (MLR). RF regression has been studied in the context of $NO_2$ co-location calibrations before, with very promising results

(Zimmerman et al., 2018). Equally for $NO_2$, but not for PM10, different linear versions of GPR have been tested by De Vito et al. (2018) and Malings et al. (2019). To the best of our knowledge, we are the first to test Ridge regression both for $NO_2$ and PM10 and GPR for PM10. Finally, we also investigate well-known issues concerning site-transferability (Masson et al., 2015; Fang and Bate, 2017; Hagan et al., 2018; Malings et al., 2019), i.e. if a calibration through co-location at one location gives rise to reliable measurements at a different location.

A key motivation for our study is the potential of low-cost sensors (costs on the order of £10-£100) to transform the level of availability of air pollution measurements. Installation costs of state-of-the-art measurement stations typically range between £10,000 and £100,000 per site, and those already high costs are further exacerbated through subsequent maintenance and calibration requirements (Mead et al., 2013; Lewis et al., 2016; Castell et al., 2017). Lower measurement costs would allow for the deployment of denser air pollution sensor networks; for portable devices possibly even at the exposure level of individuals

(Mead et al., 2013). A central complication is the sensitivity of sensors to environmental conditions such as temperature and relative humidity (Masson et al., 2015; Spinelle et al., 2015; Jiao et al., 2016; Lewis et al., 2016; Spinelle et al., 2017; Castell et al., 2017), or to cross-sensitivities with other gases (e.g. nitrogen oxide), which can significantly impede their measurement performance (Mead et al., 2013; Popoola et al., 2016; Rai and Kumar, 2018; Lewis et al., 2018; Liu et al., 2019). Low-cost sensors thus require, in the same way as many other measurement devices, sophisticated calibration techniques. Machine

learning regressions have seen increased use in this context due to their ability to calibrate for many simultaneous, non-linear dependencies. These dependencies, in turn, can for example be assessed in - relatively - expensive laboratory settings which, however, also do not always perform well in the field (Castell et al., 2017; Zimmerman et al., 2018). Here, we test the performance of sensor calibrations based on co-location measurements with established reference stations (e.g. Masson et al.,





2015; Spinelle et al., 2015; Esposito et al., 2016; Lewis et al., 2016; Cross et al., 2017; Hagan et al., 2018; Casey and Hannigan,

2018; Casey et al., 2019; De Vito et al., 2018, 2019; Zimmerman et al., 2018; Casey et al., 2019; Munir et al., 2019; Malings

et al., 2019, 2020). If sufficiently successful, these methods could help to substantially reduce the overall costs and simplify

the process of calibrating low-cost sensors.

Another challenge in relation to co-location calibration procedures is 'site-transferability'. This term refers to the measurement performance implications of moving a calibrated device from one location (where the calibration was conducted) to

another location. Some significant performance losses after site transfers have been reported (e.g. Fang and Bate, 2017; Casey

and Hannigan, 2018; Hagler et al., 2018; Vikram et al., 2019), with reasons typically not being straightforward to assign. A key

driver might be that often devices are calibrated in an environment not representative of situations found in later measurement

locations. As we discuss in greater detail below, for machine learning-based calibrations this behaviour can, to a degree, be

fairly intuitively explained by the fact that they do not tend to perform well when extrapolating beyond their training domain.

As we will show, this issue can easily occur in situations where already calibrated sensors have to measure pollution levels

well beyond the range of values encountered in their training environment.

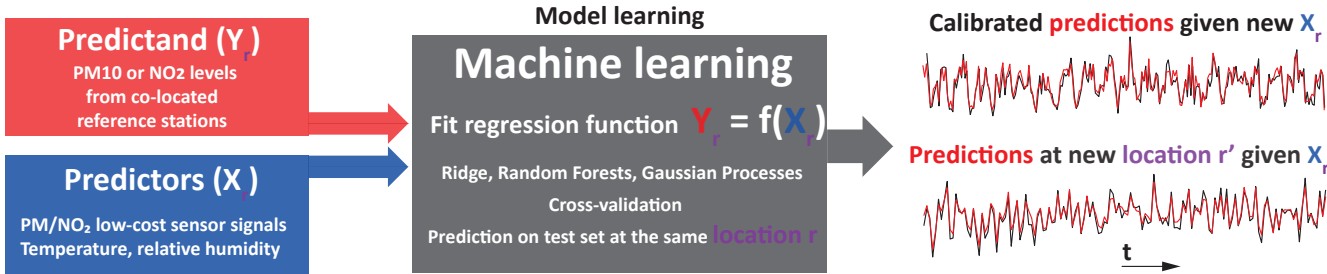

**Figure 1.** Sketch of the co-location calibration methodology. We co-locate several low-cost sensors for PM (for various particle sizes) and $NO_2$ with higher cost reference measurement stations for PM10 and $NO_2$. The low-cost sensors also measure relative humidity and temperature as key environmental variables that can interfere with the sensor signals, and for $NO_2$ calibrations we further include nitrogen oxide (NO) sensors. We formulate the calibration task as a regression problem in which the low-cost sensor signals and the environmental variables are the predictors ($X_r$), and the reference station signal the predictand ($Y_r$), both measured at the same location $r$. The time resolution is set to hourly averages to match publicly available reference data. We train separate calibration functions for each $NO_2$ and PM10 sensor, and compare three different machine learning algorithms (Ridge, Random Forest and Gaussian Process regression) with Multiple Linear Regression (MLR) in terms of their respective calibration performances. The performance is evaluated on out-of-sample test data, i.e. on data not used during training. Once trained and cross-validated, we use these calibration functions to predict PM10 and $NO_2$ concentrations given new low-cost measurements $X$, either measured at the same location $r$, or at a new location $r'$. The latter is to test the feasibility and impacts of changing measurement sites post-calibration. The time series (right) are for illustration purposes only.

We highlight that, in particular concerning the performance of low-cost PM10 sensors, a huge gap in the scientific literature

has been identified regarding issues related to co-location calibrations (Rai and Kumar, 2018). We therefore expect that our

study will provide novel insights into the effects of different calibration techniques on sensor performances, and another data





sample that other measurement studies from academia and industry can compare their results against. We will mainly use the $R^2$-score (coefficient of determination, 1 - residual sum of squares/total sum of squares) and root mean squared error (RMSE) as metrics to evaluate our calibration results, which are widely used and should thus facilitate intercomparisons. To provide a reference for calibration results perceived as 'good' for PM10, we point towards a sensor comparison by Rai and Kumar (2018) who found that low-cost sensors generally displayed moderate to excellent linearity (r2 > 0.5) across various

calibration settings. The sensors typically perform particularly well (r2 > 0.8) when tested in idealised laboratory conditions. However, their performance is generally lower in field deployments (see also Lewis et al., 2018). For PM10, this performance deterioration was, inter alia, attributed to changing conditions of particle composition, particle sizes, and environmental factors such as humidity and temperature, which are thus important factors to account for in our calibrations.

    The structure of our manuscript is as follows. In sect. 2, we introduce the low-cost sensor hardware used, the reference

measurement sources, the three measurement site characteristics and measurement periods, the four calibration regression methods, as well as the calibration settings (e.g. measured signals used) for $NO_2$ and PM10. In sect. 3, we first introduce multi-sensor calibration results for $NO_2$ at a single site, depending on the sensor signals included in the calibrations and the number of training samples used to train the regressions. This is followed by a discussion of single-site PM10 calibration results before we test the feasibility and challenges of site transfers. We discuss our results and draw conclusions in sect. 4.

## 2   Methods and Data

### 2.1   Sensor hardware

Depending on the measurement location, we deployed one set or several sets of air pollution sensors, and we refer to each set as multi-sensor 'node'. Each of these nodes consists of multiple electrochemical and metal oxide sensors for PM and $NO_2$, as well as sensors for environmental quantities and other chemical species known for potential interferences with their sensor

signals (required for calibration). For $NO_2$, we incorporated three different types of sensors in our set-up for which purchasing prices differed by an order of magnitude. One aspect of our study will therefore be to evaluate the performance gained by using the more expensive, but still relatively low-cost, sensor types. Of course, our results will only be validated in the context of our specific calibration method so that more general conclusions have to be drawn with care.

    Each multi-sensor node contained:

– Two MiCS-2714 $NO_2$ sensors produced by SGX Sensortech. These are the cheapest measurement devices deployed in our set with market costs of approximately £5 per sensor.

    – Two plantowers of the PMS5003T series PM particle sensors, which measures particles of various size categories including PM10 based on laser scattering using Mie theory. We note that particle composition does play a role in any PM calibration process as for example organic materials tend to absorb a higher proportion of incident light as compared to

inorganic materials (Rai and Kumar, 2018). Below we therefore effectively make the assumption that we measure and calibrate within composition-wise similar environments. By taking into account various particle size measures in the





calibration, we likely do indirectly account for some aspects of composition though because, to a degree, particle sizes might be correlated with particle composition. Each PMS tower also contains a temperature and relative humidity sensor and these variables were also included in our calibrations. The minimum distinguishable particle diameter for the PMS devices is 0.3 $\mu$m. The market cost is £20 per one sensor.

– An NO2-A43F 4-Electrode $NO_2$ sensor produced by Alphasense. Market cost £45.

– An NO2-B43F 4-Electrode $NO_2$ sensor produced by Alphasense. Market cost £45

– An NO-A4 4-Electrode nitric oxide sensor produced by Alphasense to calibrate against the sometimes significant interference of $NO_2$ signals with NO. Market cost £45.

– An OX-A431 4-Electrode Oxidising Gas Sensor measuring a combined signal from ozone and $NO_2$ also produced by Alphasense. We used this signal to calibrate against possible interference of electrochemical $NO_2$ measurements by ozone. Market cost £45.

– A separate temperature sensor built into the Alphasense set. It is needed to monitor the warm-up phase of the sensors.

In normal operation mode, each node provided measurements around every 30 seconds. These signals were time-averaged to hourly values for calibration against hourly public reference measurements.

## 2.2 Measurement sites and reference monitors

We conducted measurements at three sites in the Greater London area during distinct multi-week periods (Table 1). Two of the sites are located in the London Borough of Croydon, which we label CR7 and CR9 according to their UK postcodes . The third site is located in the car park of the company Alphasense in Essex, hereafter referred to as site 'CarPark' (Figure 2a). At CR7, the sensor nodes were located kerbside on a medium busy street with two lanes in either direction. At CR9, the nodes were located on a traffic island in the middle of a very busy road with three lanes in either direction (Figure 2b). For the two Croydon sites, reference measurements were obtained from co-located, publicly available measurements of London's Air Quality Network (LAQN, www.londonair.org.uk). In the two locations in question, the LAQN used Chemiluminescence detectors for $NO_2$, and ThermoScientific tapered element oscillating microbalance (TEOM) continuous ambient particulate monitors, with a Volatile Correction Model (Green et al., 2009) for PM10 measurements at CR9. For the CarPark site, PM10 measurements were conducted with a Palas Fidas Optical Particle Counter AFL-W07A. These CarPark reference measurements were provided at 15 minutes intervals. For consistency, these measurements were averaged to hourly values to match the measurement frequency of publicly available data at the other two sites.





| Site | Max. co-location period | Reference sensors | Sensor IDs |
|---|---|---|---|
| CR7 | 22 Oct 2018 - 5 Dec 2018 | NO$_2$ (London Air Quality Network) | 3-7, 11, 13-24, 27, 28 |
| CR9 | 24 Sep 2019 - 19 Jan 2020 | NO$_2$ and PM10 (London Air Quality Network) | 19 |
| CarPark | 29 Jan 2019 - 26 Apr 2019 | PM10 (Palas Fidas Optical Particle Counter AFL-W07A) | 19, 25, 26 |

**Table 1.** Overview of the measurement sites and the corresponding maximum co-location periods, which varies for each specific sensor node. Note that reference measurements for NO$_2$ and PM10 are only available for two of the three sites each. Sensors that were co-located for at least 820 active measurement hours are identified by their sensor IDs in the last column. Further note that the only sensor used to measure at multiple sites is sensor 19, which is therefore used to test the feasibility of site transfers.



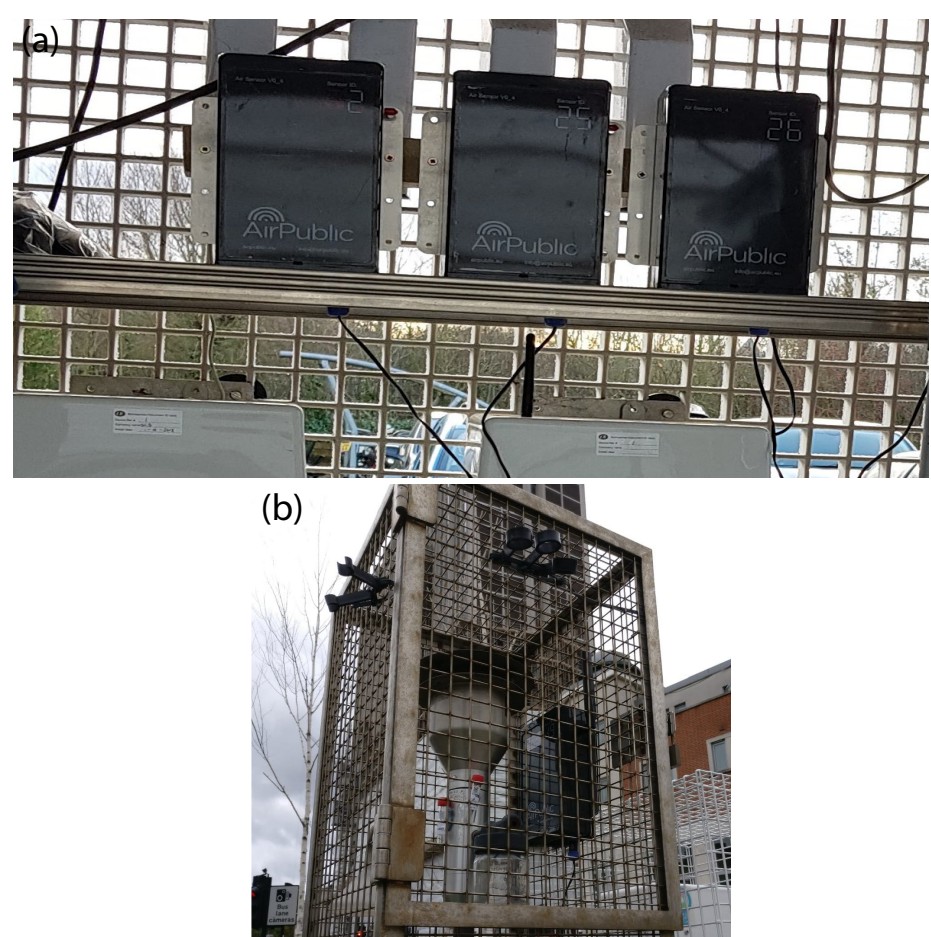

**Figure 2.** Examples of the co-location set-up of the AirPublic low-cost sensor nodes with reference measurement stations at sites (a) CarPark and (b) CR9, respectively.

## 2.3 Co-location set-up and calibration variables

In total, we co-located up to 30 nodes, labelled by identifiers (IDs) 1 to 30. For our $NO_2$ measurements, we considered the following 15 sensor signals per node to be important for the calibration process: the NO sensor (plus its baseline signal to remove noise), the $NO_2 + O_3$ sensor (plus baseline), the two intermediate cost $NO_2$ sensors (NO2-A43F, NO2-B43F) plus their respective baselines, the two cheaper MICS sensors, three different temperature sensors and two relative humidity sensors. All 15 signals can be used for calibration against the reference measurements obtained with the co-located higher cost measurement

devices. We discuss the relative importance of the different signals, e.g. the relative importance of the different $NO_2$ sensors or the influence of temperature and humidity in section 3. For the PM10 calibrations, we used two devices of the same type of low-cost PM sensor, resulting in $2 \times 10$ different particle measures used in the PM10 calibrations. In addition, we included the respective sensor signals for temperature and relative humidity, providing us with in total 24 calibration signals for PM10.





### 2.4 Calibration algorithms

We evaluate four regression calibration strategies for low-cost $NO_2$ and PM10 devices, by means of co-location of the devices with the aforementioned air quality measurement reference stations. The four different regression methods - which are multiple linear regression (MLR), Ridge regression, Random Forest (RF) regression, and Gaussian Process regression (GPR) - are introduced in detail in the following subsections. As we will show in section 3, the relative skill of the calibration methods depends on the chemical species to be measured, sample size available for calibration, as well as certain user preferences.

We will additionally consider the issue of site transferability for sensor node 19, including its dependence on the calibration algorithm used. We note that we do not include the manufacturer calibration of the low-cost sensors in our comparison here mainly because we found that this method, which is a simple linear regression based on certain laboratory measurement relationships, provided us with negative $R^2$-scores when compared with reference sensors in the field. This result is in line with other studies that reported differences between sensor performances in the field and under laboratory calibrations (see e.g.

Mead et al., 2013; Lewis et al., 2018; Rai and Kumar, 2018).

#### 2.4.1 Ridge and Multiple Linear Regression

Ridge regression is a linear least squares regression augmented by $L_2$-regularization to address the bias-variance trade-off (Hoerl and Kennard, 1970; James et al., 2013; Nowack et al., 2018, 2019). Using statistical cross-validation, the regression fit is optimized by minimizing the cost function

$$J_{\text{Ridge}} = \sum_{t=1}^{n} \left( y_t - \sum_{j=1}^{p} c_j x_{j,t} \right)^2 + \alpha \sum_{j=1}^{p} c_j^2 \qquad (1)$$

over $n$ hourly reference measurements of pollutant $y$ (i.e. $NO_2$, PM10). $x_{j,t}$ are $p$ non-calibrated measurement signals from the low-cost sensors, representing signals for the pollutant itself as well as signals recorded for environmental variables (temperature, humidity) and other chemical species that might cause interference with the signal in question. The cost function (1) determines the optimization goal. Its first term is the ordinary least squares regression error, the second term term puts a penalty on too large regression coefficients and thus avoids overfitting in high-dimensional settings by nudging the regression towards

small regression coefficients $c_j$. Smaller (larger) values of the regularization coefficient $\alpha$ put weaker (stronger) constraints on the size of the coefficients, thereby favoring overfitting (high bias). We find the value for $\alpha$ through 5-fold cross-validation, i.e. each data set is split into five ordered time slices and $\alpha$ optimized by fitting regressions for large ranges of $\alpha$-values on four of the slices at a time, and then the best $\alpha$ is found by evaluating the out-of-sample prediction error on each corresponding

remaining slice using the $R^2$-score. Each slice is used once for the evaluation step. Before the training procedure, all signals are scaled to unit variance and zero mean as to ensure that all signals are weighted equally in the regression optimization, which we explain in more detail at the end of this section. Through the constraint on the regression slopes, Ridge regression can handle settings with many predictors, here calibration variables, even in the context of strong collinarity in those predictors (Dormann





et al., 2013; Nowack et al., 2018, 2019). The resulting linear regression function $f_{\text{Ridge}}$

$$\hat{y}(t) = f_{\text{Ridge}} = c_0 + \sum_{j=1}^{p} c_j x_j(t) \qquad (2)$$


provides estimates for pollutant mixing ratios $\hat{y}$ at any time $t$, i.e. a calibrated low-cost sensor signal, based on new sensor readings $x_j(t)$. $f_{\text{Ridge}}$ represents a calibration function because it is not just based on on a regression of the pollutant signal itself against the reference, but on multiple simultaneous predictors, including those representing known interfering factors.

Multiple Linear Regression (MLR) is the simple non-regularized case of Ridge, i.e. where $\alpha$ is set to nil. MLR is therefore a
good benchmark to evaluate the importance of regularization and, when compared to RF and GPR below, of non-linearity in the relationships. As MLR does not regularize its coefficients, it is expected to increasingly lose performance in settings with many (non-linear) calibration relationships. This loss of MLR performance in high-dimensional regression spaces is related to the 'curse of dimensionality' in machine learning, which expresses the observation that one requires an exponentially increasing number of samples to constrain the regression coefficients as the number of predictors is increased linearly (Bishop, 2006). We
will illustrate this phenomenon for the case of our $NO_2$ sensor calibrations below.

Finally, we note that for Ridge regression, as also for GPR described below, the predictors $x_j$ must be normalized to a common range. For Ridge this is straightforward to understand as the regression coefficients, once the predictors are normalized, provide direct measures of the importance of each predictor for the overall pollutant signal. If not normalized, the coefficients will additionally weight the relative magnitude of predictor signals, which can differ by orders of magnitude (e.g. temperature
at around 273 Kelvin, but a measurement signal for a trace gas on the order of 0.5 amplifier units). As a result, the predictors would be penalized differently through the same $\alpha$ in equation (1), which could mean that certain predictors are effectively not considered in the regressions. Here, we normalize all predictors in all regressions to zero mean and unit standard deviation according to the samples included in each training dataset.

### 2.4.2 Random Forest regression

Random Forest (RF) regression is one of the most widely used non-linear machine learning algorithms (Breiman and Friedman, 1997; Breiman, 2001), and has already found applications in air pollution sensor calibration as well as in other aspects of atmospheric chemistry (Keller and Evans, 2019; Nowack et al., 2018, 2019; Sherwen et al., 2019; Zimmerman et al., 2018; Malings et al., 2019). It follows the idea of ensemble learning where multiple machine learning models together make more reliable predictions than the individual models. Each RF regression object consists of a collection (i.e. ensemble) of graphical
tree models, which split training data by learning decision rules (Figure 3). Each of these decision trees consists of a sequence of nodes, which branch into multiple tree levels until the end of the tree (the 'leaf' level) is reached. Each leaf node contains at least one, or several samples from the training data. The average of these samples is the prediction of each tree for any measurement of predictors $x$ defining a new traversion of the tree to the given leaf node. In contrast to Ridge regression, there is more than one tunable hyperparameter to address overfitting. One of these hyperparameters is the maximum tree depth, i.e.
the maximal number of levels within each tree, as deeper trees allow for a more detailed grouping of samples. Similarly, one





can set the minimum number of samples in any leaf node. Once this minimum number is reached, the node is not further split into children nodes. Both smaller tree depth and a greater number of minimum samples in leaf nodes mitigate overfitting to the training data. Other important settings are the optimization function used to define the decision rules, and for example the number of estimators included in an ensemble, i.e. the number of trees in the forest.

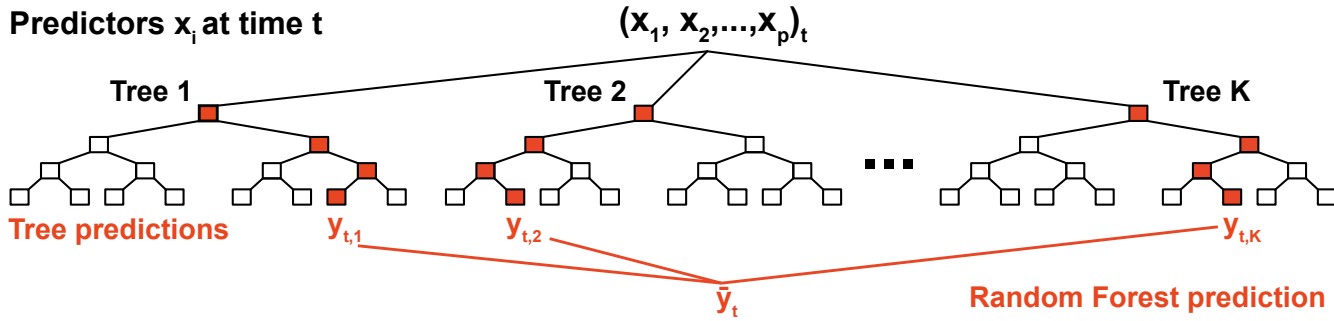

**Figure 3.** Sketch of a Random Forest regressor. Each Random Forest consists of an ensemble of $K$ trees. For visualization purposes, the trees shown here have only four levels of equal depth, but more complex structures can be learned. The lowest level contains the leaf nodes. Note that in real examples, branches can have different depths, i.e. the leaf nodes can occur at different levels of the tree hierarchy, see e.g. Figure 2 in Zimmerman et al. (2018). Once the decision rules for each node and tree are learned from training data, each tree can be presented with new sensor readings $x$ at a time $t$ to predict pollutant concentration $y_t$. The decision rules depend, inter alia, on the tree structure and random sampling through bootstrapping, which we optimize through 5-fold cross-validation. Based on the values $x_i$, each set of predictors follows routes through the trees. The training samples collected in the corresponding leaf node define the tree-specific prediction for $y$. By averaging $K$ tree-wise predictions we combat tree-specific overfitting, and finally obtain a more regularized Random Forest prediction $\bar{y}_t$.

The RF training process tunes the parameter thresholds for each binary decision tree node. By introducing randomness, for example by selecting a sub-set of samples from the training data set (bootstrapping), each tree provides a somewhat different data representation. This random element is used to obtain a better, averaged prediction over all trees in the ensemble, which is less prone to overfitting than individual regression trees. We here cross-validated the scikit-learn implementation of RF regression (Pedregosa et al., 2011) over problem-specific ranges for the minimum number of samples required to define a split

and the minimum number of samples to define a leaf node. The implementation uses an optimised version of the Classification and Regression Trees (CART) algorithm, which constructs binary decision trees using the predictor and threshold that yields the largest information gain for a split at each node. The mean squared error of samples relative to their node prediction (mean) serves as optimization criterion, as to measure the quality of a split for a given possible threshold during training. Here we consider all features when defining any new best split of the data at nodes. By increasing the number of trees in the ensemble,

the RF generalization error converges towards a lower limit. We here set the number of trees in all regression tasks to 200 as a compromise between model convergence and computational complexity (Breiman, 2001).



### 2.4.3  Gaussian Process regression

Gaussian process regression (GPR) is a widely used Bayesian machine learning method to estimate non-linear dependencies (Rasmussen and Williams, 2006; Pedregosa et al., 2011; Lewis et al., 2016; De Vito et al., 2018; Runge et al., 2019; Malings et al., 2019; Nowack et al., 2020; Mansfield et al., 2020). In GPR, the aim is to find a distribution over possible functions that fit the data. We first define a prior of possible functions that is updated according to the data using Bayes' theorem, which provides us with a posterior distribution over possible functions. The prior distribution is a Gaussian Process ($GP$)

$$Y \sim GP\left(\mu, k(x_i, x_j)\right) \tag{3}$$

with mean $\mu$ and a covariance function, or kernel, $k$ which describes the covariance between any two points $x_i$ and $x_j$. We here standard-scale (i.e. centre) our data so that $\mu$=0, meaning our GP is entirely defined by the covariance function. Being a kernel method, the performance of GPR depends strongly on the kernel (covariance function) design as it determines the shape of the prior and posterior of the Gaussian Process, and in particular the characteristics of the function we are able to learn from the data. Owing to the time-varying, continuous but also oscillating nature of air pollution sensor signals, we here use a sum kernel of a radial basis function (RBF) kernel, a white noise kernel, a Matérn kernel and a Dot-Product kernel. The RBF kernel, also known as squared exponential kernel, is defined as

$$k(x_i, x_j) = \exp\left(\frac{-d(x_i, x_j)^2}{2l^2}\right). \tag{4}$$

It is parameterized by a length scale $l > 0$, and $d$ is the Euclidean distance. The length scale determines the scale of variation in the data, and is learned during the Bayesian update, i.e. for a shorter length scale the function is more flexible. However, it also determines the extrapolation scale of the function, meaning that any extrapolation beyond the length scale is probably unreliable. RBF kernels are particularly helpful to model smooth variations in the data. The Matérn kernel is defined by

$$k(x_i, x_j) = \frac{1}{\Gamma(\nu)2^{\nu-1}}\left(\frac{\sqrt{2\nu}}{l}d(x_i, x_j)\right)^{\nu} K_\nu\left(\frac{\sqrt{2\nu}}{l}d(x_i, x_j)\right), \tag{5}$$

where $K_\nu$ is a modified Bessel function and $\Gamma$ the gamma function (Pedregosa et al., 2011). We here choose $\nu$=1.5 as the default setting for the kernel, which determines the smoothness of the function. Overall, the Matérn kernel is useful to model less smooth variations in the data than the RBF kernel. The Dot-Product kernel is parameterized by a hyperparameter $\sigma_0^2$

$$k(x_i, x_j) = \sigma_0^2 + x_i \cdot x_j \tag{6}$$

and we found that adding this kernel to the sum of kernels significantly improved our results empirically. The white noise kernel simply allows for a noise level on the data as independently and identically normally-distributed, specified through a variance parameter. This parameter is similar to, and will interact with, the $\alpha$ noise level described below, which is, however, tested systematically through cross-validation.

The Python scikit-learn implementation of the algorithm used here is based on algorithm 2.1 of Rasmussen and Williams (2006). We optimized the kernel parameters in the same way as for the other regression methods through 5-fold cross-validation, and subject to the noise $\alpha$-parameter of the scikit-learn GPR regression packages (Pedregosa et al., 2011). This





parameter is not to be confused with the $\alpha$ regularization parameter for Ridge regression and takes the role of smoothing the kernel function, as to address overfitting. It represents a value added to the diagonal of the kernel matrix during the fitting

process with larger $\alpha$ values corresponding to greater noise level in the measurements of the outputs. However, we note that there is some equivalency with the $\alpha$ parameter in Ridge as the method is effectively a form of Tikhonov regularization that is also used in Ridge regression (Pedregosa et al., 2011). Both inputs and outputs to the GPR function were standard-scaled to zero mean and unit variance based on the training data. For each GPR optimization, we chose 25 optimizer restarts with different initializations of the kernel parameters, which is necessary to approximate the best possible solution to maximize the

log-marginal likelihood of the fit. More background on GPR can be found in Rasmussen and Williams (2006).

### 2.5 Cross-validation

For all regression models, we performed 5-fold cross-validation where the data is first split into training and test sets, keeping samples ordered by time. The training data is afterwards divided into five consecutive subsets (folds) of equal length. If the training data is not divisible by five, with a residual number of samples $n$, then the first $n$ folds will contain one surplus sample

compared to the remaining folds. Each fold is used once as a validation set while the remaining four folds are used for training. The best set of model hyperparameters, or kernel functions, is found according to the average generalization error on these validation sets. After the best cross-validated hyperparameters are found, we refit the regressions models on the entire training data using these hyperparameter settings (e.g. the $\alpha$ value for which we found the best out-of-sample performance for Ridge regression).

## 3    Results

### 3.1    NO$_2$ sensor calibration

The skill of a sensor calibration function is expected to increase with sample size, i.e. the number of measurements used in the calibration process, but will also depend on aspects of the sampling environment. For co-location measurements, there will be time-dependent fluctuations in the value ranges encountered for the predictors (e.g. low-cost sensor signals, humidity,

temperature) and predictands (reference NO$_2$, PM10). The calibration range in turn affects the performance of the calibration function: if faced with values outside its training range, the function effectively has to perform an extrapolation rather than interpolation, i.e. the function is not well constrained outside its training domain. This limitation is particularly critical for non-linear machine learning functions (Nowack et al., 2018; Zimmerman et al., 2018; Hagan et al., 2018). Calibration performance will further vary for each device, even for sensors of the same make, due to unavoidable randomness in the sensor production

process (Mead et al., 2013; Castell et al., 2017). To characterize these various influences, we here test the dependence of three machine learning calibration methods, as well as of MLR, on sample size and co-location period for a number of NO$_2$ sensors.

     The NO$_2$ co-location data at CR7 is ideally suited for this purpose. 21 sensor nodes of the same make were co-located with a LAQN reference during the period October to December 2018 (Table 1). We actually co-located 30 sensor sets at the site, but





we excluded any sensors with fewer than 820 hours (samples) after outlier removal from our evaluation. The remaining sensors
measure sometimes overlapping, but different time intervals as a result of varying co-location start and end times as well as
accidental sensor malfunctions. To detect these malfunctions, and to exclude the corresponding samples, we removed outliers
(evidenced by unrealistically large measurement signals) at the original time resolution of our measurements, i.e. < 1 minute
and prior to hourly averaging. To detect outliers for removal, we used the Median Absolute Deviation (MAD) method, also
known as 'robust Z-Score method', which identifies outliers for each variable based on their univariate deviation from their
training data median. Since the median is a robust statistic to outliers itself, it is a typically a better measure to identify outliers
than for example a deviation from the mean. Accordingly, we excluded any samples $t$ from the training and test data where the
quantity

$$M_{j,t} = 0.6745 \frac{|x_{j,t} - \tilde{x}_j|}{\text{median}\{|x_{j,t} - \tilde{x}_j|\}} \tag{7}$$

takes on values >7 for any of the predictors, where $\tilde{x}_j$ is the training data median value of each predictor. To train and cross-
validate our calibration models, we took the first 820 hours measured by each sensor set and split it into 600 hours for training
and cross-validation, leaving 220 hours to measure the final skill on an out-of-sample test set. We highlight again that the test
set will cover different time intervals for different sensors, meaning that further randomness is introduced in how we measure
calibration skill. However, the relationships for each of the four calibration methods are learned from exactly the same data,
and their predictions are also evaluated on the same data, meaning that their robustness and performance can still be directly
compared. To measure calibration skill we used two standard metrics in the form of the $R^2$-score (coefficient of determination,
1 - residual sum of squares/total sum of squares) and the RMSE between the reference measurements and our calibrated signals
on the test sets. For particularly poor calibration functions, the $R^2$-score can take on infinitely negative values whereas a value
of 1 implies a perfect prediction. An $R^2$-score of 0 is equivalent to a function that predicts the correct long-term time average
of the data, but no fluctuations therein.

As discussed in section 2.1, each of AirPublic's co-location sets measures 15 signals (the predictors, or inputs) that we
consider relevant for the $NO_2$ sensor calibration against the LAQN reference signal for $NO_2$ (the predictand, or output). Each
of the 15 inputs will potentially be systematically linearly or non-linearly correlated with the output, which allows us to
learn a calibration function from the measurement data. Once we know this function, we should be able to make accurate
predictions given new inputs to reproduce the LAQN reference. As we fit two linear and two non-linear algorithms, certain
transformations of the inputs can be useful to facilitate the learning process. For example, a relationship between an input
and the output might be an exponential dependence in the original time series so that applying a logarithmic transformation
could lead to an approximately linear relationship that might be easier to learn for a linear regression function. We therefore
compared three set-ups with different sets of predictors:

    1. Using the 15 input time series as provided (label $I_{15}$).

2. Adding logarithmic transformations of the predictors ($I_{30}$).

    3. Adding logarithmic plus exponential transformations of the predictors ($I_{45}$).





which are labelled according to their total number of predictors after adding the input transformations, i.e. 15, 30 and 45. The logarithmic and exponential transformations of each input signal $A_i(t)$ are defined as

$$A_{i,\log}(t) = \log\left(A_i(t) + 1\right) \tag{8a}$$


$$A_{i,\exp}(t) = \exp\left(\frac{A_i(t)}{A_{\max} + \epsilon}\right) \tag{8b}$$

where $A_{\max}$ is the maximum value of the predictors time series and $\epsilon = 10^{-9}$. The latter prevents possible divisions by zero whereas the former prevents infinite values in the logarithmic function.

### 3.1.1 Comparison of regression models for all predictors

For a first comparison of the calibration performance of the four methods, we show $R^2$-scores and RMSEs in Table 2, rows (a) to (c), averaged across all 21 sensor nodes. GPR emerges as the best performing method for all three sets of predictor choices, reaching $R^2$-scores better than 0.8 for $I_{30}$ and $I_{45}$. This highlights that GPR should from now on be considered as an option in similar sensor calibration exercises. RF regression consistently performs worse than GPR, but slightly better than Ridge regression, which in turn outperforms MLR in all cases, but the differences are fairly small for $I_{15}$ and $I_{30}$. A notable 330 exception occurs for $I_{45}$, where the $R^2$-score for MLR suddenly drops abruptly to around 0.2. This sudden performance loss can be understood from the aforementioned 'curse of dimensionality': MLR increasingly overfits the training data as the number of predictors increases; the existing sample size becomes too small to constrain the 45 regression coefficients (Bishop, 2006; Runge et al., 2012). The machine learning methods can deal with this increase in dimensionality highly effectively and thus perform well throughout all three cases. Indeed, GPR and Ridge regression benefit slightly from the additional predictor 335 transformations. This robustness to regression dimensionality is a first central advantage of machine learning methods in sensor calibrations. Machine learning methods will be more reliable and will allow users to work in a higher dimensional calibration space compared to MLR. Having said that, for 15 input features the performance of all methods appears very similar on first sight, making MLR seemingly a viable alternative to the machine learning methods. We note, however, that there is no apparent disadvantage in using machine learning methods to prevent potential dangers of overfitting depending on sample size.



|  | Input features | MLR | Ridge | RF | GPR |
|---|---|---|---|---|---|
| (a) | $I_{15}$ | 0.74 (6.2) | 0.75 (6.1) | 0.76 (5.9) | 0.79 (5.7) |
| (b) | $I_{30}$ | 0.73 (6.3) | 0.75 (6.0) | 0.76 (5.9) | 0.81 (5.4) |
| (c) | $I_{45}$ | 0.23 (10.6) | 0.75 (6.0) | 0.76 (5.9) | 0.80 (5.5) |
| (d) | MICS, T, RH | -2.9 (28.3) | -0.03 (12.6) | 0.01 (12.2) | -0.12 (13.1) |
| (e) | A43F, T, RH | 0.25 (9.7) | 0.22 (10.1) | 0.44 (8.7) | 0.47 (8.4) |
| (f) | B43F, T, RH | 0.20 (10.6) | 0.39 (9.7) | 0.43 (9.5) | 0.49 (9.3) |
| (g) | NO/O3/B43F/T/RH | 0.68 (6.9) | 0.75 (6.2) | 0.69 (6.8) | 0.77 (6.0) |
| (h) | (g) + A43F | 0.72 (6.4) | 0.74 (6.2) | 0.75 (6.1) | 0.79 (5.7) |
| (i) | $I_{30}$ + B43F ($\tau$=1) | 0.78 (5.8) | 0.79 (5.6) | 0.78 (5.7) | 0.84 (5.0) |

**Table 2.** Average $NO_2$ sensor skill depending on the selection of predictors. Shown are average $R^2$-scores and root mean square errors (in brackets; units $\mu$g m$^{-3}$). Results are for 21 sensor sets with 600 hourly training samples each, and the evaluation is carried out for 220 test samples each. RH stands for relative humidity, T for temperature.

### 3.1.2 Calibration performance depending on sample size

We next consider the performance dependence on sample size of the training data (Figure 4). The advantages of machine learning methods become even more evident for smaller numbers of training samples, even if we consider case (b) with 30 predictors, i.e. $I_{30}$, for which we found that MLR performs fairly well if trained on 600 hours of data. The mean $R^2$-score and RMSE ($\mu$g m$^{-3}$) quickly deteriorate for smaller sample sizes for MLR, in particular below a threshold of less than 400 hours of training data. Ridge regression - its statistical learning equivalent - always outperforms MLR. In contrast, both GPR and RF regression already perform well at small samples sizes of less than 300 hours. While all methods converge towards similar performance approaching 600 hours of training data (Table 2), MLR is generally performing worse than Ridge regression and significantly worse that RF regression and GPR.

Further evidence for advantages of machine learning methods are provided in Figure 5 showing boxplots of the $R^2$-score distributions across all 21 sensor nodes depending on sample size (300, 400, 500, 600 hours) and regression method. While median sensor performances of MLR, Ridge and GPR ultimately become comparable, MLR is typically found to yield a number of poor performing calibration functions with some $R^2$-scores well below 0.6 even for 600 training hours. In contrast, the distributions are far narrower for the machine learning methods: GPR and RF do not show a single extreme outlier even after being trained on only 400 hours of data, providing strong indications that the two methods are the most reliable. After 600 hours, one can effectively expect that all sensors will provide $R^2$-scores > 0.7 if trained using GPR. Overall, this highlights again that machine learning methods will provide better average skill, but also are expected to provide more reliable calibration





functions through co-location measurements independent of sensor device and the peculiarities of the individual training and test data set.

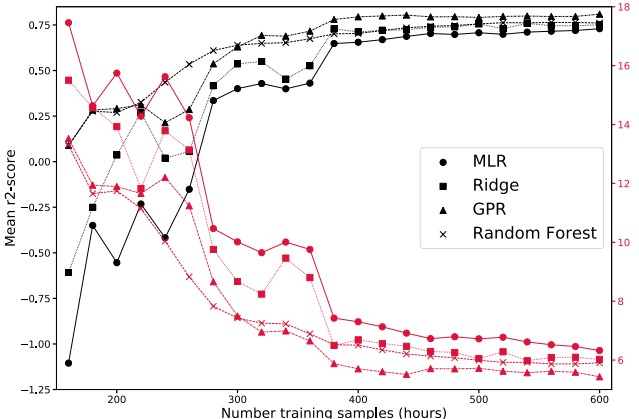

**Figure 4.** Error metrics as a function of the number of training samples for $I_{30}$, as labelled. The figure highlights the convergence of both metrics for the different regressions methods as the sample size increases. Note that MLR would not converge for $I_{45}$ owing to the curse of dimensionality. This tendency can also be seen here for small sample sizes, where MLR rapidly loses performance.





**Figure 5.** $R^2$-scores depending on calibration method and training sample size for $I_{30}$. The boxes extend from the lower to the upper quartile; inset lines mark the median. The whiskers extending from the box indicate the range excluding outliers (fliers). Each circle represents the $R^2$-score on the test set for an individual sensor (21 in total). For MLR, some sensors remain poorly calibrated even for larger sample sizes.



### 3.1.3 Calibration performance depending on predictor choices and NO$_2$ device

Tests (a) to (c) listed in Table 2 indicate that the machine learning regressions for NO$_2$, specifically GPR, can benefit slightly from additional logarithmic predictor transformations, but that adding exponential transformations on top of these predictors does not further increase predictive skill, as measured through the R$^2$-score and RMSE. Incorporating the logarithmic transformations, we next tested the importance of various predictors to achieve a certain level of calibration skill (rows (d) to (i) in Table 2). This provides two important insights: firstly, we test the predictive skill if we use the individual MICS and Alphasense

NO$_2$ sensors separately, i.e. if individual sensors are performing better than others in our calibration setting and if we need all sensors to obtain the best level of calibration performance. Secondly, we test if other environmental influences such as humidity and temperature significantly affect sensor performance.

We first tested three set-ups in which we used only the sensor signals of the two cheaper MICS devices (d) and then set-ups with the more expensive Alphasense A43F (e) and B43F (f) devices. Using just the MICS devices, the R$^2$-score drops from

0.75-0.81 for the machine learning methods to around zero, meaning that hardly any of the variation in the true NO$_2$ reference signal is captured. Using our calibration approach here, the MICS would therefore not be sufficient to achieve a meaningful measurement performance. The picture looks slightly better, albeit still far from perfect, for the individual A43F and B43F devices for which R$^2$-scores of almost 0.5 are reached using non-linear calibration methods. We note that the linear MLR and Ridge methods do not achieve the same performance, but Ridge outperforms MLR. The most recently developed Alphasense

sensor used in our study, B43F, is the best performing standalone sensor. If we add the NO/O$_3$ sensor and the humidity and temperature signals to the predictors - case (g) - its performance alone almost reaches the same as for the $I_{30}$ configuration. This implies that the interference with NO/ozone, temperature and humidity might be significant and has to be taken into account in the calibration, and that if only one sensor could be chosen for the measurements, the B43F sensor would be the best choice. By further adding the A43F sensor to the predictors the predictive skill is only mildly improved (h). Finally,

we note that, in this stationary sensor setting, further predictive skill can be gained by considering past measurement values. Here, we included the one-hour lagged signal of the best B43F sensor (i). This is clearly only possible if there is a delayed consistency, or autocorrelation, in the data, which here leads to the maximum R$^2$ generalization score of 0.84 for GPR, and related gains in terms of the RMSE. While being an interesting feature, we will not consider such set-ups in the following, because we intend sensors to be transferable among locations, and that they should only rely on live signals for the hour of

measurement in question.

In summary, using all sensor signals in combination is a robust and skilful set-up for our NO$_2$ sensor calibration and is therefore a prudent choice, at least if one of the machine learning methods is used to control for the curse of dimensionality. In particular, the B43F sensor is important to consider in the calibration, but further calibration skill is gained by also considering environmental factors and additional NO$_2$ devices.

## 3.2 PM10 sensor calibration

In the same way as for NO$_2$, we tested several calibration settings for the PM10 sensors. For this purpose, we consider the measurements for the location CarPark, where we co-located three sensors (IDs 19, 25 and 26) with a higher cost device (Table 1). However, after data cleaning, we have only 509 and 439 samples (hours) for sensors 19 and 25 available, respectively, which our NO$_2$ analysis above indicates is too short to obtain robust statistics for training and testing the sensors. Instead we focus our analysis on sensor 26 for which there are 1314 hours of measurements available. We split this data into 400 samples for training and cross-validation, leaving 914 samples for testing the sensor calibration. Below we discuss results for various calibration configurations, using the 24 predictors for PM10 (section 2.3) and the same four regression methods as for NO$_2$. The baseline case with just 24 predictors is named $I_{24}$, following the same nomenclature as for NO$_2$. $I_{48}$ and $I_{72}$ refer to the cases with additional logarithmic and exponential transformations of the predictors according to equations (8a) and (8b). In addition, we test the effects of environmental conditions, as expressed through relative humidity and temperature, by excluding these two variables from the calibration procedure, while using the $I_{48}$ set-up with the additional log-transformed predictors.

The results of these tests are summarized in Table 3. For the baseline case of 24 non-transformed predictors, RF regression (r2 = 0.70) is outperformed by Ridge regression (r2=0.79) and GPR (r2=0.79). This is mainly the result of the fact that some of the pollution values measured with sensor 26 during the test period lie outside the range of values encountered at training stage. RFs cannot predict values beyond their training range (i.e. they cannot extrapolate to higher values) and can therefore not predict those values accurately (see also Zimmerman et al., 2018; Malings et al., 2019). Instead the RF constantly predicts the maximum value encountered during training in those cases.

| | Predictors | MLR | Ridge | RF | GPR |
|---|---|---|---|---|---|
| (a) | I$_{24}$ | 0.28 (13.0) | 0.79 (6.9) | 0.70 (8.3) | 0.79 (7.1) |
| (b) | I$_{48}$ | 0.70 (8.4) | 0.80 (6.8) | 0.70 (8.3) | 0.76 (7.4) |
| (c) | I$_{72}$ | 0.47 (11.1) | 0.80 (6.8) | 0.70 (8.3) | 0.79 (7.0) |
| (d) | I$_{48}$ - RH, T | 0.76 (7.6) | 0.78 (7.2) | 0.67 (8.8) | 0.75 (7.6) |

**Table 3.** Sensor skill depending on the selection of predictors for PM10 sensor 26 at location CarPark. Shown are the R$^2$-scores and RMSEs (in brackets) evaluated over 914 hours of test data, after training and cross-validating the algorithms on 400 hours of data. RMSEs are given in $\mu$g m$^{-3}$. In (d) relative humidity (RH) and temperature (T) are removed from the calibration variables to test their importance for the measurement skill.

However, this problem is not entirely exclusive to RFs, but is inherited by all methods, with RFs only being the most prominent case. We illustrate the more general issue, which will occur in any co-location calibration setting, in Figure 6. In the training data, there are not any pollution values beyond ca. 40 $\mu$g m$^{-3}$ so that the RF predictions simply level off at that value. This is a serious constraint in actual field measurements where one would be particularly interested in episodes of highest





pollution. We note that this effect is somewhat alleviated by using GPR and even more so by Ridge regression. For the latter, this behaviour is intuitive as the linear relationships learned by Ridge will hold to a good approximation even under extrapolation to previously unseen values. However, even for Ridge regression the predictions eventually deviate from the 1:1 line for the

highest pollution levels. This aspect will be crucial to consider for any co-location calibration approach, as is also evident from the poor MLR performance, despite being another linear method. In addition, MLR sometimes predicts substantially negative values, producing an overall $R^2$-score of below 0.3, whereas the machine learning methods appear to avoid the problem of negative predictions almost entirely. In conclusion we highlight the necessity for co-location studies to ensure that maximum pollution values encountered during training and testing/deployment are as similar as possible. Extrapolations beyond 10-20

$\mu$g m$^{-3}$ appear to be unreliable even if Ridge regression is used as calibration algorithm, which is the best of our four methods to combat such extrapolation issues.

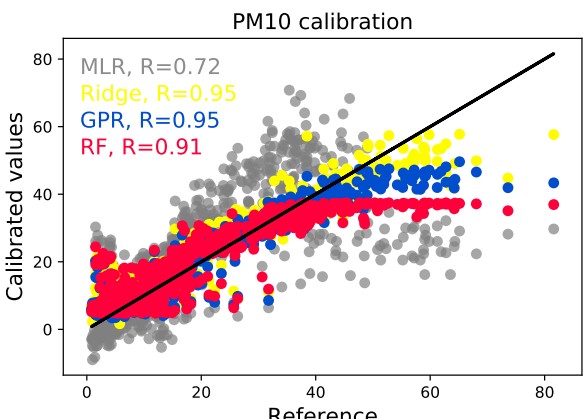

**Figure 6.** Calibrated PM10 values (in $\mu$g m$^{-3}$) versus the reference measurements for 900 hours of test data at location CarPark for the $I_{24}$ predictor set-up. The ideal 1:1 perfect prediction line is drawn in black. Inset values $R$ are the Pearson correlation coefficients.

A test with additional log-transformations ($I_{48}$) of the predictors led to test score improvements for the two linear methods (Table 3), in particular for MLR ($R^2$=0.7) but also for Ridge regression ($R^2$=0.8). This implies that the log-transformations have helped linearize certain predictor-predictand relationships. Further exp-transformations ($I_{72}$), and thus also further increasing

the predictor dimensionality, did not lead to an improvement in calibration skill. We therefore ran one final test using the $I_{48}$ set-up but without relative humidity and temperature included as predictors. This test confirmed that the sensor signals indeed experience a slight interference from humidity and temperature, at least considering the machine learning regressions. Notably, this loss of skill is not observed for MLR for which the $R^2$-score actually improves. A likely explanation for this behaviour is the curse of dimensionality that affects MLR more significantly than the three machine learning methods so that the reduction in

collinear dimensions (given the sample size constraint) is more beneficial than the information gained by including temperature and humidity in the MLR regression.



In summary, we have found that Ridge regression and GPR are the two most reliable and high-performing calibration methods for the PM10 sensor. We are able to attain very good $R^2$-scores > 0.7 for all four regression methods though. An important point to highlight is the characteristics of the training domain, in particular of the pollution levels encountered during the training data measurements. If the value range is not sufficient to cover the range of interest for future measurement campaigns, then Ridge regression might be the most robust choice to alleviate the underprediction of the most extreme pollution values. However, the power of extrapolation of any method is limited so that we underline the need to carefully check every training data set if it fulfills such crucial criteria, see also similar discussions in other calibration contexts (Hagan et al., 2018; Zimmerman et al., 2018; Malings et al., 2019).

## 3.3 Site transferability

Finally, we aim to address the question of site transferability, i.e. how reliably a sensor calibrated through co-location can be used to measure air pollution at a different location. One of the sensor nodes (ID 19) was used for $NO_2$ measurements at both locations CR7 and CR9, and was also used to measure PM10 at CR9 and CarPark, allowing us to address this question for our methodology. Note that these tests also include a shift in the time of year (Table 1), which has been hypothesized to be one potentially limiting factor in site transferability. The results of these transferability tests for PM10 (from CR9 to CarPark and vice versa) and $NO_2$ (from CR7 to CR9 and vice versa) are shown in Figures 7 and 8, respectively.

For PM10, we trained the regressions, using the $I_{24}$ predictor set-up, on 400 hours of data at either location. This emulates a situation in which, according to our results above, we limit the co-location period to a minimum number of samples required to achieve reasonable performances across all four regression methods. To mitigate issues related to extrapolation (Figure 6), we selected the last 400 hours of the time series for location CarPark (Figure 7a), and hours 600 to 1000 of the time series for location CR9 (Figure 7b), as to include the maximum possible pollution range in the training data. The maximum pollution found within the two time slices differ only by ca. $\pm$ 10 $\mu$g m$^{-3}$, for which at least Ridge regression should provide reasonable extrapolation performances. For the resulting predictions at location CarPark, using models trained on the CR9 data, we achieve generally very good $R^2$-scores ranging between 0.67 for RFs and 0.78 for Ridge regression. The site-transferred measurement performance of sensor 19 is therefore almost as good as the one for sensor 26 at the co-location site itself (Table 3), i.e. we cannot detect any significant loss in measurement performance due to the site transfer. A surprising element is that MLR performs almost as good as Ridge regression in this case, whereas it performed poorly for sensor 26, where it only achieved an $R^2$-score of 0.28 (Table 3). This underlines our previous observation that the performance of MLR is more sensitive to the specific sensor hardware, with sometimes low performance for relatively small sample sizes (Figure 5). However, our results also show again that linear methods appear to generally perform well for our PM10 sensors, with Ridge regression being the most reliable and high-performing choice overall. These results are further supported by calculations concerning the detection of only the most extreme pollution events in the time series, see also the definition of such events described in the caption of Figure 7. We characterize such events in the form of their statistical recall in our sensor measurements (the fraction of extreme pollution events in the reference time series that are also identified by our calibrated sensors) and precision (the fraction of extreme events identified by our sensors that are indeed also extreme events in the reference time series) and show the results





as inset numbers in Figures 7 and 8. Overall, these site-transfer PM10 results from CR9 to CarPark imply that sensors calibrated through co-location can achieve high measurement performance distant from the co-location site.

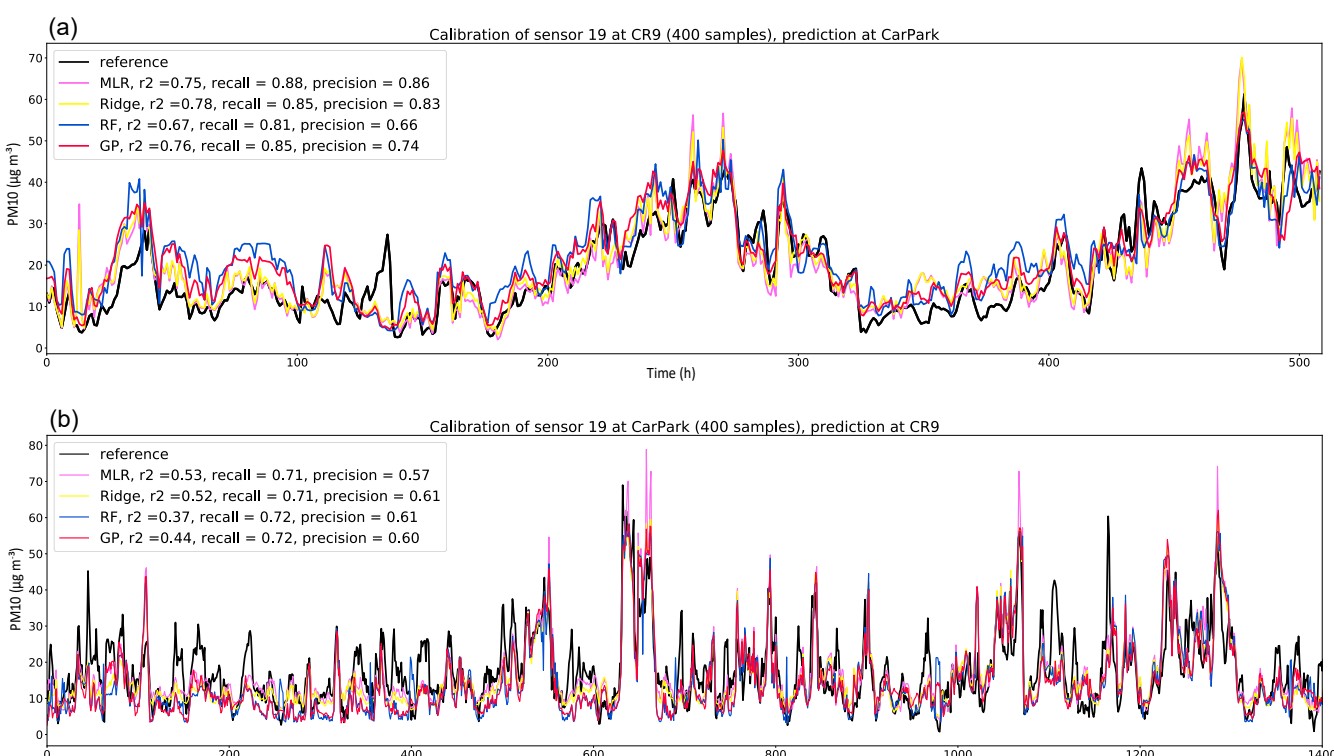

**Figure 7.** Tests of PM10 sensor site transfers using calibration models trained on 400 hours of data. (a) Predictions for the four regression models (as labelled) and reference measurements at location CarPark, using models trained at CR9, and (b) at location CR9, using models trained on data measured at CarPark. The inset values provide the $R^2$-scores for each method relative to the reference as well as the corresponding recall and precision for the detection of the strongest pollution events, which are typically of particular interest in real life situations (here defined as events when two values within the last three hours exceeded a threshold of 35 $\mu$g m$^{-3}$). For compactness, we only show data for times at which both reference and low-cost sensor data was available, and label these hours as a consecutive timeline.

However, we do find that site transferability is not always as straightforward as found for this particular case. For example, for the inverse transfer using models trained at CarPark and predicting PM10 pollution levels at CR9, we find lower $R^2$-scores
overall. There is consistency in the sense that MLR and Ridge remain the best performing methods for sensor 19 with $R^2$-scores of around 0.5, but the sensors now miss several significant pollution events in the time series. We note, however, that many of the most extreme pollution events are still detected, which is evident from the still relatively high precision and recall scores for all methods. These results underline that in general at least a good performance can be achieved with co-location calibrations, but that there are also significant challenges posed by site transfers. In particular, the problem is not necessarily
symmetric among sites, i.e. the skill of the method can depend on the direction of the transfer, even if the pollution levels at





both sites are similar. We therefore hope that our insights and results will motivate further work in this direction, with the aim to identify possible causes of such effects. We discuss some of the possible reasons for this behaviour in sect. 4.

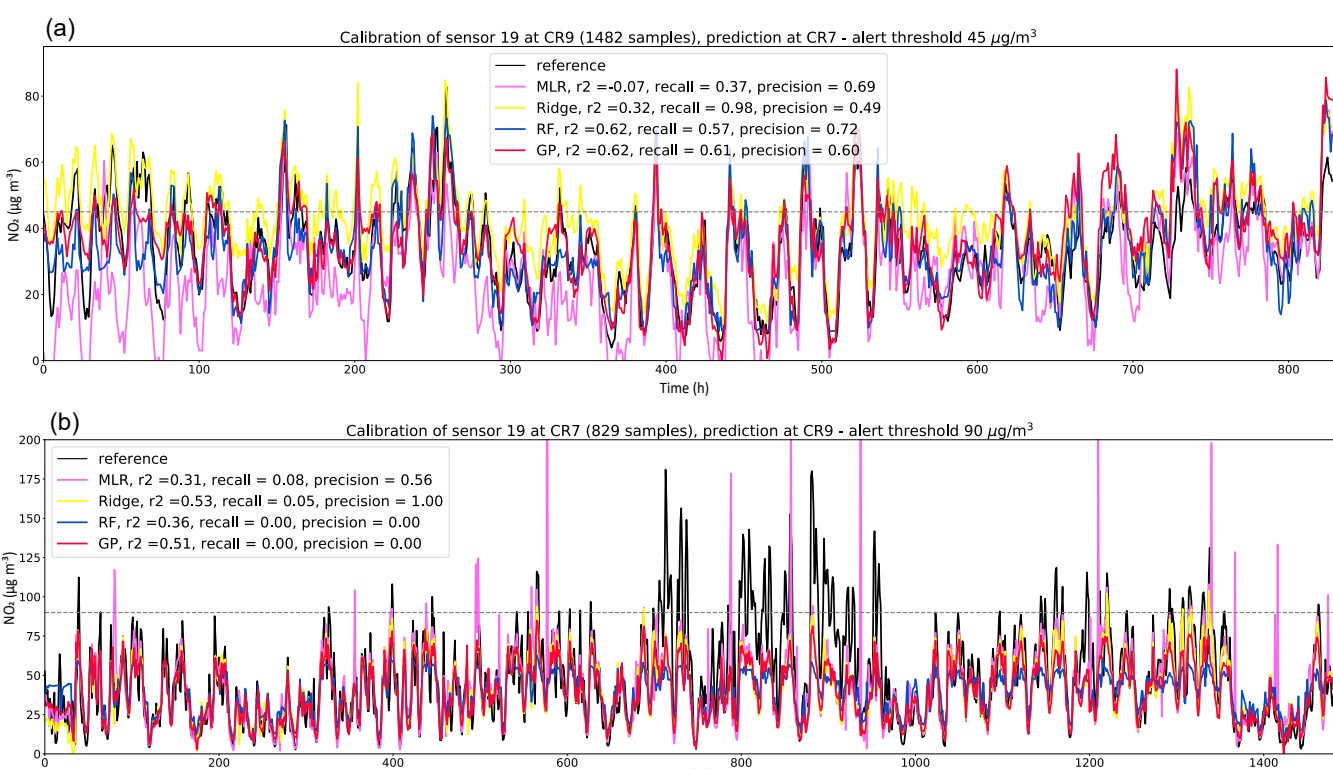

**Figure 8.** Tests of NO$_2$ sensor site transfers using calibration models trained on all available samples for locations CR7 and CR9. (a) Predictions for the four regression models (as labelled) and reference measurements at location CR7, using models trained on data from CR9, and (b) vice versa. The inset values provide the R$^2$-scores for each method relative to the reference as well as the corresponding recall and precision for the detection of the strongest pollution events, which are typically of particular interest in real life situations. Since the two locations were subject to very different pollution ranges (note the different value ranges on the y-axes), these are defined in (a) as 45 $\mu$g m$^{-3}$ and in (b) as 90 $\mu$g m$^{-3}$, and we indicate these thresholds by gray dashed lines. An extreme pollution event occurs when the threshold is exceeded for two of the last three hours. For compactness, we only show data for times at which both reference and low-cost sensor data was available, and label these hours as a consecutive timeline.

Similarly, we find promising results for the NO$_2$ sensor site transfer using the $I_{30}$ predictor set-up (Figure 8). The key challenge for the sensor transfer from CR7 to CR9 is that the maximal pollution levels at the two locations differ strongly, with peak concentration being around 100 $\mu$g m$^{-3}$ greater at CR9. To allow for the best possible learning opportunities for the regression algorithm, we therefore used all available samples for training, which are 1482 samples at CR9 and 829 samples at CR7. This leads to overall good performance of the non-linear RF and GPR methods at location CR7 using models trained at CR9. As no extrapolation is necessary, these methods achieve a good performance of R$^2$-scores > 0.6 and also a good balance





of precision and recall. The results are, however, slightly worse than for the same site calibrations (Table 2). Ridge regression

has a tendency to overpredict $NO_2$ pollution levels in this particular case, likely because it cannot capture some non-linear effects that would have limited the prediction values. As a result, it also reproduces almost all extreme pollution events where the concentration of $NO_2$ exceeds 45 $\mu$g m$^{-3}$ (recall =0.98) but also predicts many false pollution events (precision=0.49).

Despite the large sample size, MLR performs poorly for both site transfers ($R^2$=0.07 and -0.31). In particular, MLR underpredicts, sometimes providing even impossible negative pollution estimates, at CR7 whereas it provides several runaway

positive values at CR9 (Figure 8). However, at CR9 all methods struggle with the impossible challenge of extrapolation far outside their training domain, which effectively is an extreme demonstration of the effects of an ill-considered training range (cf. Figure 6). Among the machine learning methods, the effect is as expected most prominent for RF regression (r2=0.36), which cannot predict any pollution values beyond those encountered at training stage. This is a serious limitation and means that the method scores nil on precision and recall of any extreme pollution events at CR9 where $NO_2$ levels exceeded 90 $\mu$g m$^{-3}$. GPR

is slightly better at extrapolating beyond its training domain (compare also Figure 6) but still not good enough to reproduce any of the extreme pollution events, giving rise to equally low precision and recall. Ridge regression, as a regularized linear method, performs best in the sense that it is able to reproduce at least a few of the extreme events (recall=0.05) and predicting no false extreme events (precision=1.0), while still achieving an $R^2$-score of 0.53. Nonetheless, it is clear from the time series in Figure 8 that none of the regression methods works for this site transfer, simply because of the too large extrapolation range.

## 4   Discussion and conclusions

We have compared four different regressions methods to calibrate a number of low-cost $NO_2$ and PM10 sensors against reference measurement signals, by means of co-location at three separate sites in London, UK. Comparing the four regression methods our main conclusions are:

1. For the 21 $NO_2$ sensors, Gaussian Process regression (GPR) is generally performing best at the same measurement

site, followed by Ridge regression, Random Forest (RF) regression, and Multiple Linear regression (MLR). For a single sensor PM10 calibration, we find that Ridge regression and GPR attain about the same measurement performance. We note that in particular the relative performance of GPR differs greatly from a recent study by Malings et al. (2019), likely due to our different choice of kernel design.

2. Special care must be taken of the calibration conditions, in particular if sensors are thereafter used for measurements

in areas where higher pollution levels are to be expected. The linear Ridge method can best mitigate the catastrophic measurement failure in such extrapolation settings, but also fails if measurement signals deviate by more than around 10-20 $\mu$g m$^{-3}$ from the maximum pollution level in the training data. For our $NO_2$ measurement with site transfer, we find that the non-linear methods GPR and RF outperform Ridge, assuming that the training pollution range encapsulates the range of values encountered at the new site. For the PM10 sensor calibrations and corresponding site transfers, we

find that Ridge regression is the highest performing and most reliable calibration algorithm overall.





3. All three machine learning methods (Ridge, GPR, and RF regression) generally outperform, or perform as least as good as, MLR. The machine learning methods are also more reliable if many signals are used for calibration, or if the number of measurement samples is relatively small. MLR suffers most significantly from the curse of dimensionality in those settings and can produce highly erroneous results.

4. Under careful consideration of the calibration conditions given expected measurement conditions, the low-cost sensors typically achieve high performances with $R^2$-scores often exceeding 0.8 on new unseen test data.

On another note, we highlight that we sometimes found significant signals in our test data sets that were not reproduced by our low-cost sensor nodes, see e.g. the pollution peak at $t \approx 140$ in Figure 7a, even if the measured pollution value lies well within the training data range. It is hard to assign reasons to this surprising sensor behaviour, as our low-cost sensors are apparently
able to capture most of the other pollution spikes well for the same dataset. One possible reason is a calibration blind spot, i.e. we encounter a new type of sensor interference which we did not find in the training data. However, we think that this is unlikely, given that the behaviour is not found frequently, at least in this particular time series. Two other options are (a) imperfect co-location, for example we might have missed an important local pollution plume by chance, or (b) temporary sensor failures that were removed by the MAD outlier removal, i.e. that our sensors were temporarily inoperational at the time
of a pollution spike that dominated the values for the given measurement hour. We hope that future measurement campaigns can provide further insights into such calibration challenges, and hope that our study can motivate further work in this direction.

In conclusion, our results underline the potential of machine learning algorithms for the calibration of co-located low-cost NO$_2$ and PM10 sensors. At the same time, we highlight several significant challenges that will always have to be considered in similar calibration processes. For example, this includes the need for a well-adjusted calibration dataset to avoid calibration
failure if the algorithm needs to extrapolate to higher pollution values, and the role of individual choices relating to the combination of calibration variables and calibration algorithms, e.g. concerning the curse of dimensionality, predictor transformations and linearity in the predictor-predictand relationships. Recent studies indicate that the issues related to extrapolation can be mitigated through the application of hybrid models in which non-linear machine learning models are used within the training domain and a simpler linear regression approach otherwise (Hagan et al., 2018; Malings et al., 2019).We note that in particular
Ridge regression could be a good compromise, which does not require a somewhat arbitrary hybrid-model definition. Having said that, we also found that even high-dimensional linear methods have ultimately limited extrapolation skill (Figure 6) so that the consideration of the training data pollution range remains of fundamental importance. We hope that such insights will contribute to ever less expensive and more spatially dense measurements of air pollution in the future, and that our work will motivate additional measurement campaigns, testing of other calibration algorithms, and further low-cost sensor development.



*Code availability.* Analysis code will be made available on Peer Nowack's Github website.

*Data availability.* Data is available from H.G. and L.K.

*Author contributions.* PN wrote the paper and carried out the analysis, supported by LK. The study was designed and suggested by HG, LK and PN. LK, JC and HG (through AirPublic) measured and prepared the data.

*Competing interests.* The authors declare no competing interests.

*Acknowledgements.* Peer Nowack is supported through an Imperial College Research Fellowship. AirPublic is supported by ClimateKIC, DigitalCatapult and Future Cities Catapult via Organicity. AirPublic has received funding from the European Union's Horizon 2020 research and innovation programme under grant agreement No 645198.





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
