# Peer review of "Machine learning calibration of low-cost NO2 and PM10 sensors: non-linear algorithms and their impact on site transferability"

_Atmospheric Measurement Techniques, 2020_

## Referee Comment (RC1) · Carl Malings (Referee) · 28 Dec 2020

General Comments

This paper presents an application of machine learning techniques to the calibration of data from low-cost sensors. It particularly focuses on (1) the effects of different subsets and combinations of inputs, including transformed inputs, on the resulting calibration performance, (2) the comparison of regularized Ridge regression to un-regularized regression, and of Gaussian Process regression to the more common Random Forest approach, and (3) the transferability of performance between locations, especially in cases where there is a greater range of concentrations at one location as compared

to another. Overall, the paper presents and explains the issues well, will a good description of the motivation and methods. The discussion of the machine learning approaches in particular shows a good understanding of these techniques. The results presented are mainly in line with previous work in this area and help to highlight and provide greater context for some of these issues, in particular the question of transferability of calibrations between locations. The paper is fairly well written, and I believe it is suitable for publication, provided some steps are taken to clarify certain aspects and statements (as outlined below).

Specific Comments

Lines 1-8: This background information can probably be condensed to 1 or 2 sentences within the abstract.

Lines 18-19: For the sentence "In particular, none of the methods is able to extrapolate to pollution levels well outside those encountered at training stage.", I believe it should say "…none of the non-parametric methods…" or "…none of the non-linear methods…", since you later state that the linear Ridge regression is able to extrapolate. Alternatively, if you mean that the methods are able to extrapolate but may not do so well, I suggest phrasing that as "…none of the methods is able to extrapolate well to pollution levels…".

Line 76: I would recommend removing the "1 – residual sum of squares/total sum of squares" part of this sentence, as this is more of a calculation formula than a definition of the term. Instead, I would suggest including this as a numbered equation in your paper, e.g., in the results section.

Figure 2: I believe this is the first time "AirPublic" is mentioned in the context of the sensor nodes. I suggest that this be explicitly stated as the maker of the sensor nodes in the body of the paper where the sensor nodes are described.

Line 301: Same comment as for line 76.

Line 316: "logarithmic plus exponential" can be ambiguous, i.e., did you use both as separate inputs, or add them together? I would instead phrase this as "both logarithmic and exponential".

Lines 386-389: I would also suggest mentioning the importance of measuring potential interferents, like ozone and NO, since this seems to be indicated by your results as well and is a separate issue from the temperature and humidity effects.

Lines 449-451: It is not clear to me why dividing the data based on time in this way would guarantee the largest variability in pollutant concentrations.

Lines 510-512: You should specify whether this statement (in particulate the concentration range given) refers to NO2, PM10, or both.

Technical Corrections

Lines 18-19: I believe that "...none of the methods is able..." should be "...none of the methods are able...".

Lines 56-57: This sentence is rather grammatically complicated; I would suggest revising it and/or splitting it up into several sentences.

Lines 79-80: You refer to the "r2" (lower-case R, non-superscript 2) metric here, is this the same as the coefficient of determination?

Line 102: "plantowers" should be capitalized. You may also want to indicate that this is the manufacturer.

Line 112: Missing period.

Line 123: Extra space before period.

Table 1: "varies" should be "vary".

Lines 262-264: Data are plural.

Figure 4: Again, "r2" is used here.

Line 403: "r2" is being used again.

---

## Referee Comment (RC2) · Carl Malings (Referee) · 28 Dec 2020

Another suggestion I forgot to include:

I would suggest considering changing the title of the paper. Currently, it is very generic. You may want to focus on some specific result and allude to that in the title, for example, relating to how well these methods can be applied beyond their calibration site, or how the some techniques (linear/Ridge regression) are better suited for extrapolation beyond the training data range.
* * *

---

## Referee Comment (RC3) · Anonymous Referee #2 · 17 May 2021

This study investigates a promising calibration method for the low-cost air pollution sensors, through co-location with public measurement stations and regression towards the station reference data. Four machine learning algorithms for the regression, namely Ridge, Random Forest (RF), Gaussian Process Regression (GPR) and Multiple Linear Regression (MLR) are implemented and compared. The influence from co-location training conditions and choice of calibration input data, and the issues of "site-transferability" are also discussed. Overall, this interesting manuscript is well written, and potentially contributes to the scientific community and also the environmental protection agencies. My comments are listed below:
1. My biggest concern is about the "site-transferability". 1) Any more insights on the key factors limiting the site-transferability? Description between Line 102-110 has stated that the PMS5003T series PM particle sensors are based on laser scattering using Mie theory, and the composition-wise similar environments are assumed for measurement and calibration, which potentially limits the site-transferability. If the authors could also explain the measurement principles for other sensors and how would they potentially limit the site-transferability? 2) The use of multi-sensor node is a very good idea to involve more factors for the regression and thus potentially enhance the "site-transferability". But the description for the set-up of sensor hardware in Sect. 2 is not clear. In Line 99, "Each multi-sensor node contained", does it mean that all nodes listed in Table 1 have the same set-up of sensor hardware? If so, why the same set-up is used for NO2 and PM10, respectively? In Table 1, different nodes spanning over different measurement time? Why so many nodes are used at CR7 site for the relatively shorter time period? Please add clarification. 3) In Figs. 7-8, the results for "site-transferability" looks promising except NO2 concentrations predicted at CR9 between ∼700-1000 h are largely underestimated. What if the node-19 is calibrated at CR9, would the peak concentrations between ∼700-1000 h be properly predicted? Why the calibration is not performed at the site with a larger range for the observed concentrations (e.g., CR9 rather than CR7 for NO2 for the Sect. 3.1)? I mean, the calibration at CR7 discussed at Sect. 3.1 seems to have a limited applicability.

2. Besides the spatial variability, the temporal variability is also very important. I mean, given the fixed co-location, if the calibration performed during the winter time, is applicable to make proper predictions during the summer time? Furthermore, any special treatment for the calibration of the sensor signals during occasionary condition (e.g., intensive washout during the heavy rain, dust event, or strong temperature inversion, etc.) when the observed concentrations at the reference site are extraordinarily high or low?

3. In Table 2, it is very interesting that the performance is greatly enhanced if the sensor

signals relevant with NO/O3 are involved into regression (i.e., (g) v.s. (f)), and further in-volvement of more factors does not seem to improve the performance significantly. Any more detailed explanation (the description between Line 375-378 lacks of insights)? How about the performance for the NO/O3/T/RH, if it is better than A43F/T/RH and B43F/T/RH? And the computing cost for each selection of predictors?

4. It would be very nice if the authors could use a table to summarize and compare the key assumptions, algorithms, advantages/disadvantages, computing cost etc. for the four regression methods.

5. In Fig. 5, how the circles (for each individual node) are calculated, as each node has a different time span? Table 2 and Fig. 4 are based on the average of the 21 nodes, right? Please add clarification.

6. Why PM10 rather than PM2.5 is focused on? PM2.5 is more relevant with the public health issues.

---

## Author Comment (AC1) · 24 Jun 2021

**Dr Peer Nowack**

[Figure]

**University Lecturer**
Atmospheric Chemistry and Data Science
Climatic Research Unit
School of Environmental Sciences
University of East Anglia
Norwich, NR4 7TJ
United Kingdom
p.nowack@uea.ac.uk

**Imperial College Research Fellow**
Grantham Institute
Department of Physics
Data Science Institute
Imperial College London
London, SW7 2AZ
United Kingdom
p.nowack@imperial.ac.uk

24th June 2021

**Atmospheric Measurement Techniques Manuscript amt-2020-473**

We thank Dr Malings for the overall positive feedback and helpful comments, which have helped to further improve our manuscript. We address each of his comments (*italic font*) point-by-point below.

**Reply to Reviewer #1:**

*This paper presents an application of machine learning techniques to the calibration of data from low-cost sensors. It particularly focuses on (1) the effects of different subsets and combinations of inputs, including transformed inputs, on the resulting calibration performance, (2) the comparison of regularized Ridge regression to un-regularized regression, and of Gaussian Process regression to the more common Random Forest approach, and (3) the transferability of performance between locations, especially in cases where there is a greater range of concentrations at one location as compared to another. Overall, the paper presents and explains the issues well, will a good description of the motivation and methods. The discussion of the machine learning approaches in particular shows a good understanding of these techniques. The results presented are mainly in line with previous work in this area and help to highlight and provide greater context for some of these issues, in particular the question of transferability of calibrations between locations. The paper is fairly well written, and I believe it is suitable for publication, provided some steps are taken to clarify certain aspects and statements (as outlined below).*

*Specific Comments*

*Lines 1-8: This background information can probably be condensed to 1 or 2 sentences within the abstract.*

**We agree. We have shortened this text to:**

**"Low-cost air pollution sensors often fail to attain sufficient performance compared with state-of-the-art measurement stations, and typically require expensive laboratory-based calibration procedures. A repeatedly proposed strategy to overcome these limitations is calibration through co-location with public measurement stations."**

*Lines 18-19: For the sentence "In particular, none of the methods is able to extrapolate to pollution levels well outside those encountered at training stage.", I believe it should say ". . .none of the non-parametric methods. . ." or ". . .none of the non-linear methods. . .", since you later state that the linear Ridge regression is able to extrapolate. Alternatively, if you mean that the methods are able to extrapolate but may not do so well, I suggest phrasing that as ". . .none of the methods is able to extrapolate well to pollution levels. . .".*

We can see how this statement could be misunderstood. Indeed, Ridge regression can extrapolate - to a degree - outside the training range, but is ultimately also limited in this sense. In other words, Ridge can also not extrapolate "well outside" the training range. Gaussian Process regression is also better at extrapolating than are Random Forests (cf. Figure 6). We agree that our current wording is difficult to follow for someone who has not read the paper yet. To clarify this, we have rephrased to:

"We also highlight several key limitations of the machine learning methods, which will be crucial to consider in any co-location calibration. In particular, all methods are fundamentally limited in how well they can reproduce pollution levels that lie outside those encountered at training stage. We find, however, that the linear Ridge regression outperforms the non-linear methods in such extrapolation settings. GPR can allow for a small degree of extrapolation, whereas RFR can only predict values within the training range. This algorithm-dependent ability to extrapolate is one of the key limiting factors when the calibrated sensors are deployed away from the co-location site itself."

*Line 76: I would recommend removing the "1 – residual sum of squares/total sum of squares" part of this sentence, as this is more of a calculation formula than a definition of the term. Instead, I would suggest including this as a numbered equation in your paper, e.g., in the results section.*

We have removed this text in response and have added the equation to section 3.1 (page 13 of the revised manuscript).

*Figure 2: I believe this is the first time "AirPublic" is mentioned in the context of the sensor nodes. I suggest that this be explicitly stated as the maker of the sensor nodes in the body of the paper where the sensor nodes are described.*

In section 2.1, we now write:

"Depending on the measurement location, we deployed one set or several sets of air pollution sensors, and we refer to each set (provided by London-based AirPublic Ltd) as multi-sensor 'node'."

*Line 301: Same comment as for line 76.*

Done.

*Line 316: "logarithmic plus exponential" can be ambiguous, i.e., did you use both as separate inputs, or add them together? I would instead phrase this as "both logarithmic and exponential".*

Thank you for pointing out. We have changed the text accordingly.

*Lines 386-389: I would also suggest mentioning the importance of measuring potential interferents, like ozone and NO, since this seems to be indicated by your results as well and is a separate issue from the temperature and humidity effects.*

We agree and have changed to paragraph to:

"In summary, using all sensor signals in combination is a robust and skilful set-up for our $NO_2$ sensor calibration and is therefore a prudent choice, at least if one of the machine learning methods is used to control for the curse of dimensionality. In particular, the B43F sensor is important to consider in the calibration, but further calibration skill is gained by also considering

**environmental factors, the presence of interference from ozone and NO, and additional NO₂ devices.”**

*Lines 449-451: It is not clear to me why dividing the data based on time in this way would guarantee the largest variability in pollutant concentrations.*

**The chosen sub-periods contain the highest PM10 concentrations across the total measurement periods for either location. For Figure 7a, the Min-Max possible range is well reflected and for location CR9 (Figure 7b), we sample both the maximum (at around 650 hours) and close to minimum values (at around 980 hours). However, it is true that slightly different choices could have been made (i.e. where to center the 400-hour period). We therefore agree that it is better to change the text to:**

**“To mitigate issues related to extrapolation (Figure 6), we selected the last 400 hours of the time series for location CarPark (Figure 7a), and hours 600 to 1000 of the time series for location CR9 (Figure 7b). This way we still emulate a possible minimal scenario of 400 consecutive hours of co-location while also including near maximum and minimum pollution values within our training data (given the available measurement data). We note that alternative sampling approaches, such as random sampling with shuffling of the data, could lead to artificial effects at validation and testing stage, because of autocorrelation effects that could inflate apparent calibration skill.”**

**Where we now also highlight disadvantages of e.g. data shuffling and random sampling.**

*Lines 510-512: You should specify whether this statement (in particular the concentration range given) refers to NO2, PM10, or both.*

**Done.**

*Another suggestion I forgot to include: I would suggest considering changing the title of the paper. Currently, it is very generic. You may want to focus on some specific result and allude to that in the title, for example, relating to how well these methods can be applied beyond their calibration site, or how the some techniques (linear/Ridge regression) are better suited for extrapolation beyond the training data range.*

**For the revised manuscript, we have changed the title to:**

**“Machine learning calibration of low-cost NO₂ and PM10 sensors: non-linear algorithms and their impact on site transferability”**

**We have also taken over all technical corrections suggested by Dr Malings.**

---

## Author Comment (AC2) · 24 Jun 2021

Our response concerning the manuscript title change is included in our reply to the first reviewer comment.

---

## Author Comment (AC3) · 24 Jun 2021

**Dr Peer Nowack**

[Figure]

**University Lecturer**
Atmospheric Chemistry and Data Science
Climatic Research Unit
School of Environmental Sciences
University of East Anglia
Norwich, NR4 7TJ
United Kingdom
p.nowack@uea.ac.uk

**Imperial College Research Fellow**
Grantham Institute
Department of Physics
Data Science Institute
Imperial College London
London, SW7 2AZ
United Kingdom
p.nowack@imperial.ac.uk

24th June 2021

**Atmospheric Measurement Techniques Manuscript amt-2020-473**

**Reply to Reviewer #2:**

**We thank the anonymous reviewer for the overall positive feedback and helpful comments, which have helped to further improve our manuscript. We address each comment (*re-stated in italic font*) point-by-point below.**

*This study investigates a promising calibration method for the low-cost air pollution sensors, through co-location with public measurement stations and regression towards the station reference data. Four machine learning algorithms for the regression, namely Ridge, Random Forest (RF), Gaussian Process Regression (GPR) and Multiple Linear Regression (MLR) are implemented and compared. The influence from co-location training conditions and choice of calibration input data, and the issues of "site-transferability" are also discussed. Overall, this interesting manuscript is well written, and potentially contributes to the scientific community and also the environmental protection agencies.*

*My comments are listed below:*

*1. My biggest concern is about the "site-transferability".*

*1) Any more insights on the key factors limiting the site-transferability? Description between Line 102-110 has stated that the PMS5003T series PM particle sensors are based on laser scattering using Mie theory, and the composition-wise similar environments are assumed for measurement and calibration, which potentially limits the site-transferability. If the authors could also explain the measurement principles for other sensors and how would they potentially limit the site-transferability?*

**Thank you for this important comment. We agree that there are a number of potential reasons for why side transferability can be limited. In our manuscript we aim to highlight the importance of the choice of machine learning calibration functions and aspects such as the training range of values sampled over, the number of training samples, and if the regression algorithm is linear or not. We reflect this more clearly in our revised manuscript, in particular through the change in manuscript title in response to the Reviewer #1.**

**In terms of the measurement principle, we think that the sensors are widely used and we have not produced the sensors ourselves and have not got the data to compare how different measurement principles could have affected our results. Any discussion would thus be highly speculative and could not be of quantitative nature, and is thus beyond the scope of our manuscript in our opinion. However, we agree that the fact that measurement principles could also affect transferability outcomes is worth highlighting. We have therefore added a comment**

to the discussion and conclusions (lines 533-544) on potential other factors that could also affect side transferability:

"One possible reason is a calibration blind spot, i.e. we encounter a new type of sensor interference which we did not find in the training data. For example, this could be substantial changes in environmental conditions, or e.g. PM particle composition, which are not captured by the calibration function. In our interpretation of results, this would represent again an extrapolation with respect to the predictors and/or the predictands. […] Interesting aspects to explore as part of future work might be to compare how sensitive site transfer performances are to the measurement principle of the $NO_2$ and PM10 devices used, and effects of the seasonal cycle (e.g. winter-based calibrations to be used during summer). We hope that future measurement campaigns can provide further insights into such calibration challenges, and hope that our study can motivate further work in this direction."

In addition, we had already mentioned studies that studied such factors in the introduction where we note that (lines 62-68):

"Some significant performance losses after site transfers have been reported (Fang and Bate, 2017; Casey and Hannigan, 2018; Hagler et al., 2018; Vikram et al., 2019}, with reasons typically not being straightforward to assign. A key driver might be that often devices are calibrated in an environment not representative of situations found in later measurement locations. As we discuss in greater detail below, for machine learning-based calibrations this behaviour can, to a degree, be fairly intuitively explained by the fact that they do not tend to perform well when extrapolating beyond their training domain. As we will show, this issue can easily occur in situations where already calibrated sensors have to measure pollution levels well beyond the range of values encountered in their training environment."

*2) The use of multi-sensor nodes is a very good idea to involve more factors for the regression and thus potentially enhance the "site-transferability". But the description for the set-up of sensor hardware in Sect. 2 is not clear. In Line 99, "Each multi-sensor node contained", does it mean that all nodes listed in Table 1 have the same set-up of sensor hardware? If so, why the same set-up is used for NO2 and PM10, respectively? In Table 1, different nodes spanning over different measurement time? Why so many nodes are used at CR7 site for the relatively shorter time period? Please add clarification.*

Yes, we indeed used the same set-up for each sensor node and each node contained measurements for $NO_2$, various particle sizes etc. This was mainly motivated by how we operate such sensors in real world settings where we usually want to measure as many pollutants as possible simultaneously. Of course, there will be variance in the production of sensors of the same make, so that the sensor nodes are not expected to behave identically.

The different measurement periods were caused by random sensor-drop outs, sensor availability (e.g. delayed delivery, used elsewhere). We simply tried to use many sensors as possible at any given time during the period in which the sensors were run. In conclusion, these choices were all motivated by rather practical boundary conditions which we did not have full control over.

To clarify these points, we have adapted the text to the following (line 102):

"Each multi-sensor node contained (i.e. all nodes consist of the same types of individual sensors):"

We have further added the following explanation to the beginning of section 2.1 (page 4) on sensor hardware:

"Each node thus allows for simultaneous measurement of multiple air pollutants, but we will focus on individual calibrations for $NO_2$ and $PM10$ here because these species were of particular interest to our own measurement campaigns. We note that other species such as $PM2.5$ are also included in the measured set of variables. We will make our low-cost sensor data available (see Data Availability statement), which will allow other users to test similar calibration procedures for other variables of interest (e.g. $PM2.5$, ozone). A caveat is that appropriate co-location data from higher cost reference measurements might not always be available."

In addition, we have clarified in the caption of Table 1 (page 6):

"Overview of the measurement sites and the corresponding maximum co-location periods, which vary for each specific sensor node and co-location site due to practical aspects such as sensor availability and random occurrences of sensor failures.

*3) In Figs. 7-8, the results for "site-transferability" looks promising except NO2 concentrations predicted at CR9 between ~700-1000 h are largely underestimated. What if the node-19 is calibrated at CR9, would the peak concentrations between ~700-1000 h be properly predicted? Why the calibration is not performed at the site with a larger range for the observed concentrations (e.g., CR9 rather than CR7 for NO2 for the Sect. 3.1)? I mean, the calibration at CR7 discussed at Sect. 3.1 seems to have a limited applicability.*

The idea behind these figures is to demonstrate limitations of site transferability (i.e. to train on data measured at one site and to predict on the other), especially to highlight extrapolation issues if the two sites have very different pollution ranges. Predictions of the kind Reviewer #2 suggests here – i.e. at the same site - generally perform well, especially if the training range contains the test range of pollution values, see e.g. $R^2$-scores in Table 2. Therefore, if node 19 was calibrated at CR9 it would be possible to predict the peak concentrations between 700-1000h better. However, this would only be the case if at least one or two of the peaks would also be contained in the training dataset as otherwise e.g. RFR would not be able to extrapolate to such high values.

*2. Besides the spatial variability, the temporal variability is also very important. I mean, given the fixed co-location, if the calibration performed during the winter time, is applicable to make proper predictions during the summer time? Furthermore, any special treatment for the calibration of the sensor signals during occasionary condition (e.g., intensive washout during the heavy rain, dust event, or strong temperature inversion, etc.) when the observed concentrations at the reference site are extraordinarily high or low?*

Changing background conditions would in our interpretation of the results represent an extrapolation task, because the new conditions would not have been contained in the training data. Each co-location measurement campaign thus has to be aware of the potential implications of training on non-representative training conditions, which is indeed a key point we are trying to make in our paper. Given our measurement periods (Table 1), we sample some degree of seasonal variability but, of course, such tests could be extended. We have thus highlighted this point in the revised conclusions and discussion section of our manuscript (lines 533-537):

"One possible reason is a calibration blind spot, i.e. we encounter a new type of sensor interference which we did not find in the training data. For example, this could be substantial changes in environmental conditions, or e.g. PM particle composition, which are not captured by the calibration function. In our interpretation of results, this would represent again an extrapolation with respect to the predictors and/or the predictands."

In the same paragraph, we now mention that the seasonal cycle could be interesting to study, too:

**"Interesting aspects to explore as part of future work might be to compare how sensitive site transfer performances are to the measurement principle of the NO$_2$ and PM10 devices used, and effects of the seasonal cycle (e.g. winter-based calibrations to be used during summer)."**

*3. In Table 2, it is very interesting that the performance is greatly enhanced if the sensor signals relevant with NO/O3 are involved into regression (i.e., (g) v.s. (f)), and further involvement of more factors does not seem to improve the performance significantly. Any more detailed explanation (the description between Line 375-378 lacks of insights)? How about the performance for the NO/O3/T/RH, if it is better than A43F/T/RH and B43F/T/RH? And the computing cost for each selection of predictors?*

**It is intuitive and well-known that NO$_2$ sensors can suffer from interference with NO/ozone. It seems that this is supported by our results and this is the point we discuss here. Using a set-up without any NO$_2$ sensors but only NO/ozone sensors would not make a lot of sense to us, especially in terms of robustness, given that we ultimately want to measure NO$_2$. The computing cost of selecting additional predictors is in our opinion negligible (orders of a few seconds to minutes at most on a typical laptop nowadays). The discussion on possible choices concerning NO$_2$ sensor selection is motivated by the actual cost of purchasing additional devices, i.e. how much the inclusion of additional sensor signals actually helps in improving sensor performances and which degree of measurement accuracy could be reached with a reduced set-up.**

*4. It would be very nice if the authors could use a table to summarize and compare the key assumptions, algorithms, advantages/disadvantages, computing cost etc. for the four regression methods.*

**This is indeed a very good idea – thank you for the suggestion. We now include a fourth table in the manuscript, discussed in the conclusions section (page 25), which summarizes a few key points. For compactness, we have left out the computing cost because all calculations can fairly easily be carried out on standard computing systems/laptops nowadays and we believe that the focus should thus be on actual calibration performances.**

*5. In Fig. 5, how the circles (for each individual node) are calculated, as each node has a different time span? Table 2 and Fig. 4 are based on the average of the 21 nodes, right? Please add clarification*

**These are test set R$^2$-scores (220 hours for each node) for each individual node and the numbers in Table 2 and Figure 4 are indeed average values. We discuss the split of training and test data in section 3.1:**

**"To train and cross-validate our calibration models, we took the first 820 hours measured by each sensor set and split it into 600 hours for training and cross-validation, leaving 220 hours to measure the final skill on an out-of-sample test set. We highlight again that the test set will cover different time intervals for different sensors, meaning that further randomness is introduced in how we measure calibration skill. However, the relationships for each of the four calibration methods are learned from exactly the same data, and their predictions are also evaluated on the same data, meaning that their robustness and performance can still be directly compared. To measure calibration skill we used two standard metrics in the form of the R$^2$-score (coefficient of determination) and the RMSE between the reference measurements and our calibrated signals on the test sets. For particularly poor calibration functions, the R$^2$-score can take on infinitely negative values whereas a value of 1 implies a perfect prediction. An R$^2$-score of 0 is equivalent to a function that predicts the correct long-term time average of the data, but no fluctuations therein."**

We have added a short clarification to the caption of Figure 5 as well:

"Node-specific $I_{30}$ $R^2$-scores depending on calibration method and training sample size, evaluated on consistent 220-hour test datasets in each case (see main text)."

We have further added the following clarifying sentence to the captions of Table 2 and Figure 4:

"Results are averaged over the 21 low-cost sensor nodes with 600 hourly training samples each, and the evaluation is carried out for 220 test samples each."

*6. Why PM10 rather than PM2.5 is focused on? PM2.5 is more relevant with the public health issues.*

Arguably PM2.5 is a very important metric, but both PM10 and PM2.5 are frequently used in policy and scientific domains. We chose PM10 here for project-specific reasons (our collaborators were specifically interested in PM10), but we will make our PM2.5 data public so that other researchers could investigate PM2.5, too, which is beyond the scope of our own study. We have added an explanation to our revised manuscript (see also reply to another comment above):

"We note that other species such as PM2.5 are also included in the measured set of variables. We will make our low-cost sensor data available (see Data Availability statement), which will allow other users to test similar calibration procedures for other variables of interest (e.g. PM2.5, ozone). However, we note that appropriate co-location data from higher cost reference measurements might not always be available."